# LEARNING SYSTEM DYNAMICS FROM SENSORY INPUT UNDER OPTIMAL CONTROL PRINCIPLES

## ABSTRACT

Identifying the underlying dynamics of actuated physical systems from sensory input is of high interest in control, robotics, and engineering in general. In the context of control problems, existing approaches decouple the construction of the feature space where the dynamics identification process occurs from the target control tasks, potentially leading to a mismatch between feature and state spaces: the systems may not be controllable in feature space, and synthesized controls may not be applicable in state space. Borrowing from the Koopman formalism, we propose instead to learn an embedding of both the states and controls into a feature space where the dynamics are linear, and include the target control task in the learning objective in the form of a differentiable and robust optimal control problem. We validate the proposed approach with simulation experiments using systems with non-linear dynamics, demonstrating that the controls obtained in feature space can be used to drive the corresponding physical systems and that the learned model can serve for future state prediction.

## 1 INTRODUCTION

The study of dynamical systems is a key element in understanding many physical phenomena. Such systems are typically ruled by ordinary differential equations over state variables that contain enough information to describe and determine their behavior, and analytical models of these systems are traditionally derived as solutions of the differential equations in question. However, it is hard to fully model mathematically most real-life phenomena: they may have very complex dynamics with constantly changing interactions with the environment, and the state of the physical systems involved may be unknown or not fully observable. On the other hand, the physical systems themselves, if not their internal states, can be observed, with sensory data providing implicit information about the underlying (and unknown) states. Sensory data is used in a wide range of applications to represent the state of systems from past measurements in the form of feature spaces (Brunton et al., 2016b; Arbabi et al., 2018; Bruder et al., 2019; Brunton et al., 2021). These models are of high practical interest since they enable building compact representations compared to the density of measurements (e.g., images). They also enable lifting the state of the system to a higher dimensional space where predictive models can be built. However, even when effective, the estimated models and feature spaces remain highly uninterpretable, and using them in solving control problems remains challenging. Linear models on the contrary are easily interpretable, and enable exact and effective control when coupled with linear optimal control solvers like the linear quadratic regulator (LQR) (Liberzon, 2011). In particular, Koopman operator theory (Koopman, 1931) has gained a lot of interest recently (Brunton et al., 2016b; Arbabi et al., 2018; Brunton et al., 2021; Proctor et al., 2016; Abraham et al., 2017; Morton et al., 2018; Korda & Mezić, 2018). It guarantees the existence of a linear (if typically infinite-dimensional) representation of the dynamics of the observables (vector-valued functions) defined over the state space. Finite dimensional approximations have been proposed, and dynamic mode decomposition (DMD) (Schmid, 2010) is of particular interest in this context. In DMD, an approximation of the Perron-Frobenius operator (the adjoint to the Koopman operator) is constructed in the form of a matrix representing the transition from one observation to the next. Proctor et al. in Proctor et al. (2016) first extended the use of DMD to actuated systems and modeled the system dynamics as a linear function of the state representation and the control. Several works have built upon this approximation (Morton et al., 2018; Li et al., 2020) and various methods for estimating the corresponding operators have been proposed (Morton et al., 2018; Li et al.,

2020; Xiao et al., 2021). In all these works, the operators are constructed to solve a prediction task, assuming the controls are known and, once obtained, they are used in a control task, typically an LQR problem. However, there are two main issues with this decoupled approach. First, the learned features may not be adapted to the control task, since they have not been trained for it. Thus, the modeled dynamics are not guaranteed to be effective when used as (linear) constraints to minimize a given (quadratic) cost. We believe that including the control problem in the learning process should help learning features that are well suited for both prediction *and* control. Second, when looking for a couple of matrices that satisfy the desired linearity property in the feature space, it is assumed that the dynamics are linear in the real controls. This is a strong assumption that is not necessarily satisfied, and is not justified by Koopman operator theory. In addition, the dimension of the observation representations is usually larger than that of the (unknown) states, resulting in a representation that has potentially more degrees of freedom than the real system. Thus, if the dimension of the control in feature space remains that of the original system, the system might not be controllable anymore, in the sense of the Kalman criterion for controllability introduced in Kalman (1964). This is more likely to happen when the observation and control feature spaces have similar dimensions. To address this issue, we propose to also lift the controls to a higher dimensional space in order to avoid rank deficiency issues in the controllability matrix.

Our contribution is threefold. First, we learn a representation space of non-linear dynamical systems where their dynamics are forced to be linear and can be controlled effectively. We enable this by including a control task in the representation learning framework and by lifting the controls to a higher dimensional space, reducing the gap between the number of degrees of freedom and the control dimension in this space. Finally, we validate our approach on controlling effectively in simulation pendulum and cartpole systems with various physical parameters directly from raw images.

## 2 RELATED WORK

**Data-driven approaches for learning linear dynamics.** Data-driven approaches based on Koopman operator theory (Koopman, 1931) have gained in popularity in the context of building dynamical models directly from measurements (Mezić & Banaszuk, 2004; Mezić, 2005). Borrowing from the Perron-Frobenius operator, adjoint to the Koopman operator (Brunton et al., 2021; Klus et al., 2015), these approaches are grounded in the formalism enabled by the dynamic mode decomposition (DMD) (Schmid, 2010), where a matrix models the system dynamics. In this context, several approaches have been proposed to construct a representation space with linear dynamics. Arbabi et al. (2018), Bruder et al. (2019), Abraham et al. (2017), and Brunton et al. (2016a) handcraft the so-called basis functions on pre-defined observable functions on the state space, exploiting domain-specific knowledge. Such basis functions are effective when studying a single system, but they do not generalize well to other systems by construction. Morton et al. (2018), Li et al. (2020), Xiao et al. (2021), Takeishi et al. (2017), Yeung et al. (2019), Lusch et al. (2018), Azencot et al. (2020) and Xiao et al. (2023) propose a more generic approach which consists in learning the basis functions from either the states themselves or from partial measurements of them (obtained through sensors which act as observables). In these works, the DMD matrix is either learned jointly with the basis functions (Xiao et al., 2021; Yeung et al., 2019; Lusch et al., 2018; Azencot et al., 2020; Xiao et al., 2023), or computed as the best least-squares fit to a given optimization problem (Morton et al., 2018; Li et al., 2020; Takeishi et al., 2017).

**Linear dynamics and control.** The DMD framework (Schmid, 2010) has also been extended to actuated systems for model identification and control purposes in Proctor et al. (2016). Building upon this extension, Brunton et al. (2016b), Arbabi et al. (2018), Bruder et al. (2019), Morton et al. (2018), Li et al. (2020) and Bounou et al. (2021) seek to identify both dynamics and control matrices, either using learned representation spaces or handcrafted ones, and then use the obtained matrices to solve control problems. In a different line of work, Watter et al. (2015) seek to identify a time-dependent locally linear model of a system's dynamics in a learned embedding space.

**Control learning.** In all the mentioned approaches, the control task is decoupled from the representation space construction, potentially leading to a mismatch between the representation space and the state space. Shi & Meng (2022) partially address this problem by learning an encoding space for controls on top of learning one for the state inputs. Yin et al. (2022) also address this by including an LQR control task in the learning framework. Particularly, they introduce a framework where a controller is learned along with an embedding space using the assumed known states. This idea of

learning a controller was also explored in Rückert et al. (2013) where the authors learn the system dynamics and intrinsic cost function parameters in an RL setting. In this paper, we also include an LQR control task in the learning framework in order to learn a dynamical system representation which is forced to be controllable directly. We notably leverage differentiable optimization techniques from Amos et al. (2018), Vien & Neumann (2021), Roulet & Harchaoui (2022) and Bounou et al. (2023) in order to include a differentiable LQR controller in the learning framework. Contrary to Yin et al. (2022) and Shi & Meng (2022), we only use measurements (images) of the states to learn the representation space and build the dynamics models and assume the states themselves are unknown. This is a key difference between our approaches.

## 3 METHOD

### 3.1 CONTROLLED DYNAMICAL SYSTEMS

In this work, we study discrete time actuated dynamical systems, i.e., systems whose discrete state $x_t$ in $\mathcal{X} \subset \mathbb{R}^n$ follows an equation of the form

$$x_{t+1} = f_t(x_t, u_t), \tag{1}$$

where $u_t$ in $\mathcal{U} \subset \mathbb{R}^p$ is a control input vector. In practical settings, the model $f_t$ is unknown, and the states $x_t$ are unknown or only partially observable through sensors $g_t$. In this work, we consider sensor measurements $d_t = g_t(x_t)$ (e.g., images), and seek to learn encodings of the unknown states of the system directly from these measurements.

### 3.2 APPROACH

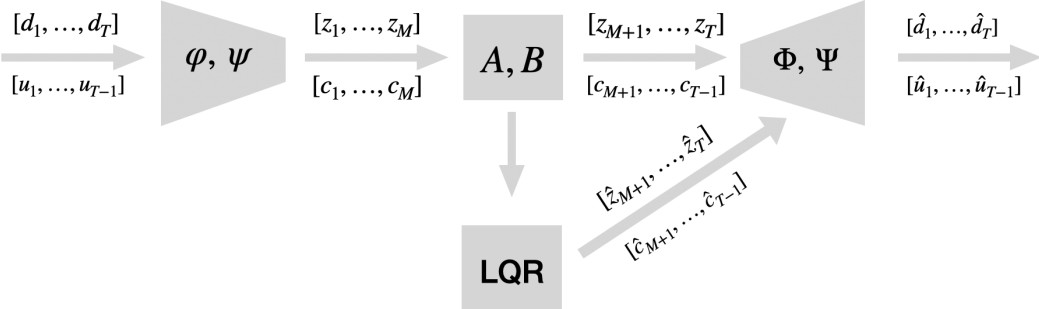

Figure 1: Method: Measurements $d_t$ and controls $u_t$ are input respectively to the image and control encoders. The first $M$ elements are encoded and used to estimate dynamics matrices $A$ and $B$. Codes from $M + 1$ to $T$ are obtained in two ways: (1) as solutions to the LQR problem of driving the system from its configuration at time $M$ to its configuration at time $T$ in $T - M$ time steps, and (2) as predictions obtained with the forward linear model associated with the matrices $(A, B)$. All codes $z_t$ and $c_t$ are then input respectively to the image and control decoders, and image and control reconstructions $\hat{d}_t$ and $\hat{u}_t$ are obtained.

We use an autoencoder to learn embeddings of both the states and the controls that have three properties: first, the code for the system state at time $t$, constructed from system measurements at that time $t$, should contain enough information to capture the behavior of the system at that time. Second, we want the dynamics to be linear in code space, even though they may be arbitrary in the original state space. This is inspired in part by the Koopman linear representation of arbitrary dynamics for non-actuated systems. Third, we want the system to be controllable in the learned representation space. Because of the first and second properties, the system dynamics are lifted to a higher dimensional space. In such a space, the system representation has more degrees of freedom, and might not be controllable with the original controls anymore. To cope with this, we propose to also learn an encoding of the controls through a second autoencoder, in order to lift the controls to a higher dimensional space.

**Encoding.** Let us consider a dynamical system governed by Eq. (1). We assume that the states $x_t$ and the model $g_t$ are unknown, and that we only have access to a sequence of $T$ measurements $d_t$ in $\mathcal{I} \subset \mathbb{R}^{C \times H \times W}$ of the system (images in our case). We want to learn the parameters of encoders $\varphi_\theta : \mathcal{I} \to \mathbb{R}^n$ and $\psi_\theta : \mathcal{U} \to \mathbb{R}^d$ such that:

$$z_t = \varphi_\theta(d_t), \ \ c_t = \psi_\theta(u_t), \ \ \text{for } t = 1, \ldots, T. \tag{2}$$

The encoders $\varphi_\theta$ and $\psi_\theta$ are learned in an auto-encoding fashion jointly with their decoder counterparts $\Phi_\theta : \mathbb{R}^n \to \mathcal{I}$ and $\Psi_\theta : \mathbb{R}^d \to \mathcal{U}$ such that $\hat{d}_t = \Phi_\theta(z_t)$ and $\hat{u}_t = \Psi_\theta(c_t)$.

**Linear model estimation.** Inspired by Koopman operator theory, we want the dynamics to be linear in the representation space. We use an approach similar to the DMD approximation to estimate dynamics and control matrices $A$ and $B$. Our goal is to learn a representation space where the dynamics are linear, i.e., we want to find matrices $A$ and $B$ such that

$$z_{t+1} = Az_t + Bc_t \ \ \text{for all } t. \tag{3}$$

To do so, we split the sequence of $T$ measurements in two subsequences. The first one, from $1$ to $M$ (where $M$ is an integer smaller than $T$), is used to estimate the matrices $A$ and $B$, and the second one, from $M+1$ to $T$, is used to verify that these matrices enable prediction. Formally, $A$ and $B$ are estimated using $z_1, \ldots, z_M$ and $c_1, \ldots, c_M$ by solving the linear least-squares problem

$$\min_{(A,B)} \sum_{t=1}^{M-1} \|z_{t+1} - Az_t - Bc_t\|_2^2. \tag{4}$$

To solve this problem, we follow Li et al. (2020) and Bounou et al. (2021): defining the matrices $Z_1 = [z_1 \ldots z_{M-1}]$, $Z_2 = [z_2 \ldots z_M]$ and $C = [c_1 \ldots c_{M-1}]$, we solve

$$(A, B) = \arg\min_{(V,W)} \|Z_2 - VZ_1 - WC\|_F^2. \tag{5}$$

Because the data points $z_t$, $c_t$ are learned encodings, the linear least-squares problem (5) may be poorly conditioned. To avoid instabilities during training and ensure we always obtain a solution, we use the proximal method of multipliers (Rockafellar, 1976) and solve iteratively the well-posed problem

$$(A^{k+1}, B^{k+1}) = \arg\min_{(A,B)} \|Z_2 - AZ_1 - BC\|_F^2 + \frac{\rho}{2}\|A - A^k\|_F^2 + \frac{\rho}{2}\|B - B^k\|_F^2 \tag{6}$$

until convergence, starting from random initial matrices $A^0$ and $B^0$. Unlike other regularization schemes such as adding L2 regularizations to $A$ and $B$, this iterative procedure is guaranteed to converge to a solution to the original problem (5) (Rockafellar, 1976; Parikh & Boyd, 2014), and not to a solution to a shifted problem as may happen with L2 regularizations.

**Future representation prediction.** The future code values $z_{M+1}, \ldots z_T$ are obtained using the forward linear model:

$$z_{t+1} = Az_t + Bc_t, \ \ \text{for } t \text{ in } M, \ldots, T-1. \tag{7}$$

This step ensures that the learned encoders $\varphi_\theta, \psi_\theta$ and the matrices $A$ and $B$ are suited for prediction.

**LQR control task.** Finally, we want the system to be controllable in code space. The goal is to drive the system from an initial configuration in the encoding space to a target in that same space. We formalize this as an LQR problem:

$$(\hat{z}, \hat{c}) = \arg\min_{z,c} \sum_{t=M}^{T-1} [(z_t - z_T^*)^T Q_t (z_t - z_T^*) + c_t^T R_t c_t] + (z_T - z_T^*)^T Q_f (z_T - z_T^*) \tag{8a}$$

$$\text{s.t. } z_{t+1} = Az_t + Bc_t, \ z_M = \varphi(d_M) \text{ and } z_T^* = \varphi(d_T), \tag{8b}$$

where $Q_t$ and $Q_f$ are symmetric positive semi-definite state cost matrices, and $R_t$ is a symmetric positive definite control cost matrix. Here, $z_M$ is the initial configuration of the system, and $z_T^*$ is the target configuration we want the system to reach. Both encode measurements of the system in initial and target configurations (e.g., if the system is a pendulum and the measurements are images, $z_M$ and $z_T^*$ are encodings of the images of the pendulum at the original position and at the desired

target position.). The quadratic cost in Eq. (8a) has a tracking term that pushes the variables $z_t$ to be close to the target $z_T^*$ all along the control trajectory.

**Supervision.** Equations (2), (7) and (8) lead to three situations: from 1 to $M$, the variables $z_t$ are obtained directly by encoding the corresponding measurements $d_1, \ldots, d_M$. From $M+1$ to $T$, they are obtained by the forward linear model $z_{t+1} = A z_t + B c_t$, using encodings of the known controls $u_{M+1}, \ldots, u_T$. Finally, $\hat{z}_{M+1}, \ldots, \hat{z}_T$ are obtained by solving the LQR problem of driving the system in the encoding space from $\hat{z}_M$ to the target $\hat{z}_T$. There are two types of control features: control encodings $c_1, \ldots c_M$ from Eq. (2), and solutions to the LQR problem $\hat{c}_{M+1}, \ldots, \hat{c}_T$ from problem (8). In all cases, the codes $z_t$ and $c_t$ should match the original measurements once decoded:

$$\hat{d}_t = \Phi(z_t) \quad \text{and} \quad \hat{u}_t = \Psi(c_t) \quad \text{for } t = 1, \ldots, T. \tag{9}$$

**Training objective.** Our model is trained to minimize the empirical risk of the loss:

$$\mathcal{L}_\theta(\{\mathbf{d}^i, \mathbf{u}^i\}_{i=1,\ldots,N}) = \frac{1}{N} \sum_{i=1}^{N} \Big( \underbrace{\sum_{t=1}^{M} \|d_t^i - \Phi(\varphi(d_t^i))\|_2^2}_{\text{(a) Measurements AE loss}} + \underbrace{\sum_{t=M}^{T-1} \|d_{t+1}^i - \Phi(A_i \varphi(d_t^i) - B_i \psi(u_t^i))\|_2^2}_{\text{(b) Prediction loss}}$$

$$+ \underbrace{\sum_{t=0}^{T-1} \|u_t^i - \Psi(\psi(u_t^i))\|_2^2}_{\text{(c) Controls AE loss}} + \underbrace{\sum_{t=M+1}^{T} \|d_t^i - \Phi(\hat{z}_t^i)\|_2^2}_{\text{(d) LQR reco loss}} + \underbrace{\sum_{t=M}^{T-1} \|u_t^i - \Psi(\hat{c}_t^i)\|_2^2}_{\text{(e) LQR controls loss}} \Big), \tag{10}$$

where $\theta$ is the set of parameters of the encoders $\varphi_\theta$ and $\psi_\theta$ and decoders $\Phi_\theta$ and $\Psi_\theta$. The first term uses the auto-encoder measurements loss and ensures that the encoder learns a representation with enough information about the system at a given timestep to reconstruct the observations. The second one uses a prediction loss which ensures that the estimated matrices $(A, B)$ enable the prediction of future codes in the latent space (that are then decoded to the measurement space). The third term uses the auto-encoder controls loss and ensures that the control encoding can be mapped back to its original space. The last two terms are associated with the control task: they ensure that the decodings of the optimal solutions to the LQR match the original optimal measurements and control trajectories. Since the different terms in the loss are not in the same scale nor the same unit, we add scaling coefficients before each one of them for training.

**LQR formulation with delayed coordinates.** In practice, raw measurement data may not contain sufficient information for future prediction. It is the case for example when we observe a single image of an oscillating pendulum at a given time step and we can not tell from this image whether the pendulum is going up or down. Instead, encodings of at least two consecutive measurements are needed to predict the future code. This case is easily handled exploiting the classical trick of using augmented feature representations associated with multiple frames, similarly to what is done in Takeishi et al. (2017) and Bounou et al. (2021). Let us define the augmented vector $\tilde{z}_t^{1:h} = \begin{bmatrix} z_{t-h+1}^T & \ldots & z_t^T \end{bmatrix}^T$, where $h$ is the number of consecutive measurements we choose to consider. The new dynamics equation becomes $z_{t+1} = A^{1:h} z_t^{1:h} + B c_t$, where $A^{1:h} = [A_1 \quad \ldots \quad A_h]$. To respect the formalism of Eq. (8b), we define the augmented matrices $\tilde{A}$ and $\tilde{B}$ as

$$\tilde{A} = \begin{bmatrix} 0 & I & 0 & \ldots & \ldots & 0 \\ 0 & 0 & I & 0 & \ldots & 0 \\ \ldots & \ldots & \ldots & \ldots\ldots & \ldots & \\ \ldots & \ldots & \ldots & \ldots & \ldots & I \\ A_1 & \ldots & \ldots & \ldots & \ldots & A_h \end{bmatrix}, \tilde{B} = \begin{bmatrix} 0 \\ \ldots \\ 0 \\ B \end{bmatrix}, \tag{11}$$

where $\tilde{A}$ is of size $(hn \times hn)$ and $\tilde{B}$ is of size $(hn \times d)$. We also define augmented initial and terminal constraints $\tilde{z}_M^{1:h} = \begin{bmatrix} z_{M-h+1}^T & \ldots & z_M^T \end{bmatrix}^T$ and $\tilde{z}_T^{:*} = \begin{bmatrix} z_{T-h+1}^T & \ldots & z_T^T \end{bmatrix}^T$, both of size $hn$. With these notations, the new dynamics equation becomes:

$$\tilde{z}_{t+1} = \tilde{A} \tilde{z}_t + \tilde{B} c_t. \tag{12}$$

## 4 EXPERIMENTS

### 4.1 EXPERIMENTAL SETUP

**Datasets.** We generate datasets of optimal trajectories for a pendulum and a cartpole. For each system, we specify initial and terminal conditions. We use Pinocchio (Carpentier et al., 2015–2021) to simulate geometric models of the systems and Crocoddylp Mastalli et al. (2020) to solve the optimal control problem of taking the systems from an initial position to a target position. For the pendulum, the initial and target positions are initial and target angular positions and velocities for the pole. Initial positions are sampled uniformly between $\frac{3\pi}{4}$ and $\frac{5\pi}{4}$, and target positions are sampled uniformly between $\frac{-\pi}{4}$ and $\frac{\pi}{4}$. We choose such distributions of initial and target positions to ensure the optimal control trajectories we obtain are not trivial. The cartpole we consider in our experiments is a cart translating along a horizontal axis. A pole is attached to it, and it can rotate over an axis orthogonal to both itself and the cart axis. For the cartpole, the initial and target positions are initial and target Cartesian positions and velocities for the tip of the pole. For all trajectories, the target position of the cartpole is one where the pole is up and vertical. Both datasets contain different trajectories of pendulums and cartpoles with different physical parameters. For the pendulums, the pole length is sampled uniformly between 0.5 m and 0.8 m, and the mass is sampled uniformly between 0.5 kg and 2 kg. For the cartpoles, the carts are cylinders of radius 0.1 m and of length and mass sampled uniformly respectively between 0.3 m and 0.7 m, and 2 kg and 5 kg. The pole length is fixed at 0.7 m, and its mass is sampled uniformly between 0.5 kg and 2 kg. Each trajectory is then generated from different initial and target positions. We render both generated systems into black and white videos of frames of $64 \times 64$ pixels. We train our models on 4000 videos of 4 seconds (200 frames), and test them on 1000 videos of the same duration. Systems are different in both sets: their physical parameters are diverse, and initial and target positions are random. Thus trajectories in both sets are different.

**Architecture.** The image and control autoencoders are the only functions with trainable parameters in our model (the dynamics matrices $A$ and $B$ are latent variables). The image autoencoder is made of a en encoder with 6 convolutional blocks, where each convolutional block except for the last one is composed by a $3 \times 3$ convolution layer, a max-pooling layer, a batch normalization layer and a ReLu layer. The last block does not have a ReLu layer. The decoder is symmetric to the encoder (with transposed convolutional layers instead of convolutional layers). The encoder of the control autoencoder is made of a dense layer followed by a tangent hyperbolic (tanh) non-linearity to enable having both positive and negative controls. The decoder is also symmetric to the encoder. We learn a single image autoencoder and a single control autoencoder for all the pendulum systems, and another single image autoencoder and single control autoencoder for all the cartpole systems. The main reason for training a single auto-encoder for different system parameters is to be robust to the uncertainty on the physical parameters of the system, similar to domain randomization (Mehta et al. (2020) and Valtchev & Wu (2021)).

We use the differentiable LQR solver introduced in Bounou et al. (2023). The LQR cost matrices $Q_t$ and $R_t$ are set to the identity matrix. We train our models during 200 epochs on a single Tesla V100 GPU.

### 4.2 LEARNING DYNAMICS WITH A CONTROL TASK IN THE OBJECTIVE

Given video measurements of a system performing an optimal trajectory and the associated optimal controls, we learn a representation space where the system dynamics are forced to be linear and exploit this linearity to solve LQR control problems in this space. Solutions of the LQR problem are then used to control the original system. Since the trajectories are optimal in the measurement space (by construction during the datasets generation), we also look for optimal control trajectories in the representation space. In our case, the measurements are videos, and the overall task is supervised in the following way: given sequences of length $T$ and a time index $M$ smaller than $T$, codes $z_t$ and $c_t$ from 1 to $M$ are encodings of the measurements $d_t$ and the controls $u_t$ from 1 to $M$: $z_t = \varphi(d_t)$ and $c_t = \psi(u_t)$. We then compute the codes $z_t$ from $M + 1$ to $T$ in two ways. First, we use the forward linear model $z_{t+1} = Az_t + B\psi(u_t)$ using the known $u_t$ that are encoded, and starting from $z_M = \varphi(d_M)$. This is done to ensure that the encoders $\varphi, \psi$ and the matrices $(A, B)$ are suited for prediction. Next, we solve the LQR problem associated with the task of taking the system from $M$

to $T$ (in $T - M$) time steps. This is done to ensure that the encoders $\varphi, \psi$ and the matrices $(A, B)$ are suited for control. The definition of codes from $M + 1$ to $T$ is consistent in both ways: first, the codes verify the dynamics constraints by construction; second, they are solution of the LQR problem and are thus feasible (i.e., they also verify the dynamics constraints).

### 4.3 CONTROL RESULTS

Figure 2 shows the average over 1000 test samples of the distance to the target angular position (in degrees) as a function of the control horizon (in s) for the pendulum. We see that our method, which includes the control task in the learning framework (blue), leads to a significant improvement in the control performance compared to Bounou et al. (2021) (in red) which does not include the control task. Figure 3 shows the average over 1000 test samples of the distance to target pole and cart positions for the cartpole. We show the average pole angle error in the left plot, and the average cart position error in the right plot. On this more challenging system, the improvement from our method (in blue) is even more significant. The approach of Bounou et al. (2021), which is trained without a control task (in red), fails completely.

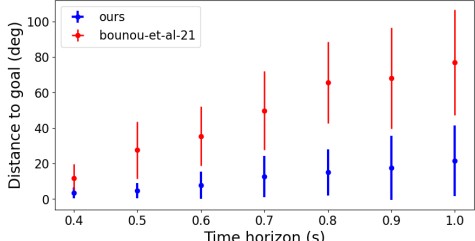

Figure 2: **Pendulum.** Average distance to target as a function of the control horizon. Model trained on multiple trajectories of a single pendulum.

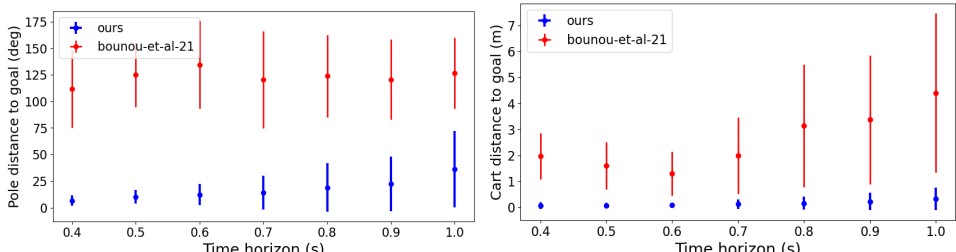

Figure 3: **Cartpole.** Average distance to target as a function of the control horizon. **Left.** Pole distance to target. **Right.** Cart distance to target.

Table 1: **Success rates for controlling a pendulum** given various thresholds and different control horizons $T_c$. An experiment is considered a success when the pole reaches its target position within a range of $\mp\alpha$ degrees. Success rates were evaluated on 1000 control tasks.

|  | Single pendulum dataset | | | Multiple pendulums dataset | | |
|---|---|---|---|---|---|---|
|  | $\alpha = 5°$ | $\alpha = 10°$ | $\alpha = 15°$ | $\alpha = 5°$ | $\alpha = 10°$ | $\alpha = 15°$ |
| $T_c = 0.4$ s | 100% | 100% | 100% | 80% | 97% | 99% |
| $T_c = 0.5$ s | 97% | 100% | 100% | 64% | 91% | 97% |
| $T_c = 0.6$ s | 90% | 99% | 99% | 46% | 72% | 87% |
| $T_c = 0.7$ s | 85% | 98% | 99% | 28% | 50% | 69% |
| $T_c = 0.8$ s | 66% | 92% | 98% | 21% | 41% | 60% |
| $T_c = 0.9$ s | 50% | 85% | 95% | 21% | 39% | 57% |
| $T_c = 1$ s | 51% | 84% | 93% | 17% | 33% | 45% |

Figures 4 and 5 show control trajectories (left) and the resulting controlled system trajectories (right in Fig 4, middle and right in Fig. 5). The control trajectories are obtained by solving the LQR problems of taking a pendulum and a cartpole from an initial position ($t = M$) to a target one ($t = T$) in 30 time steps (0.6 s). The initial and target positions are directly specified in the form of images, enabling image-based control. The ground truth controls (in green solid lines) correspond to the optimal controls in the original space. The LQR controls (in dashed-lines) are obtained by solving the LQR control problem of Eq. (8) in the encoding space, then decoding the obtained sequence back to the original space with the learned controls decoder $\Psi$. The blue dashed-lines (ours) correspond to controls obtained with a model trained with our method, for both future prediction and control, while the red dashed-lines correspond to controls obtained with the model introduced in Bounou et al. (2021) which is trained for future prediction only, i.e., the terms (d) and (e) in the training loss equation 10 are ignored. We see that in the case where the model is trained for both prediction and control (ours in blue), the decoded controls are very close to the ground truth ones. In addition, these decoded controls are effective in driving the systems to their target position. This is not the case anymore for the model of Bounou et al. (2021) which is trained for prediction only, for which the decoded controls are very different from the ground truth ones, and fail in driving the systems to their target position.

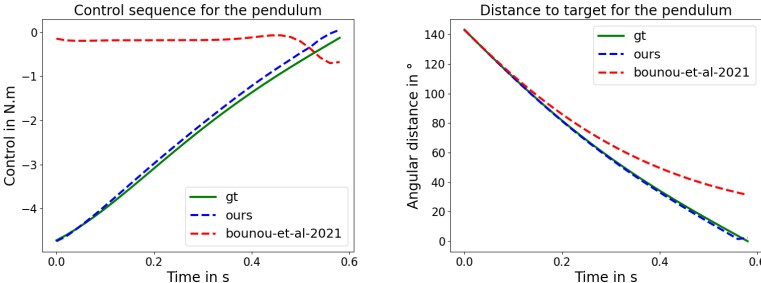

Figure 4: LQR problem of taking a pendulum from an initial position to a target position in 0.6 s (or 30 time steps.). **Left**: control trajectories. **Right**: distance to target angular position over time.

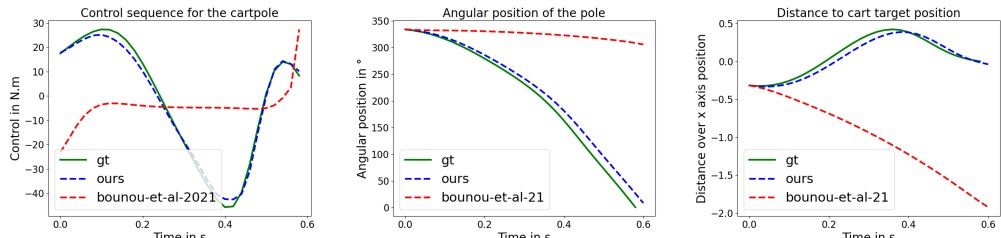

Figure 5: LQR problem of taking a cartpole system from an initial position to a target position in 0.6s (or 30 time steps.). **Left**: control trajectories. **Middle**: angular position of the pole over time. **Right**: cart distance to its target position over time.

The appendix shows additional examples, including a qualitative example of including the control task in the learning framework for the pendulum in Fig. 6 and the cartpole in Fig. 7 A qualitative example of the effectiveness of including the control task in the learning framework can be seen in Fig. 6 in the appendix for the pendulum. It is all the more noticeable in the case of the cartpole in Fig. 7 in the appendix, which is not surprising since its dynamics are more chaotic than the pendulum's.

We also report success rates of the models trained with a control task in Table 1. The model trained on a dataset with trajectories of a single pendulum has higher success rates than the model trained on a dataset with trajectories from multiple pendulums, but the latter still achieves satisfactory results since it drives the pole to less than 15° from its target position in almost all the trials for short horizons ($T_c < 0.8$ s), and in almost half the trials for longer horizons.

Table 2: MPC. Success rates of driving the pole of the pendulum to $\mp\alpha$ degrees within different MPC time horizons.

|          | $\alpha = 5°$ | $\alpha = 10°$ | $\alpha = 15°$ |
|----------|---------------|----------------|----------------|
| MPC 4s   | 34%           | 37%            | 39%            |
| MPC 8s   | 58%           | 60%            | 61%            |
| MPC 12s  | 73%           | 75%            | 76%            |
| MPC 16s  | 78%           | 80%            | 81%            |

**Model predictive control.** In practice, accurate open-loop control over long horizons is of low interest. Instead, it is common to take advantage of sensor feedback and update the control trajectory when new measurements are acquired. A standard way of doing it is through model predictive control (MPC) (Camacho & Alba, 2013) where a control problem is solved over a given horizon and only the first control of the found sequence is applied at each step.

We have run experiments using 100 pairs of initial and target image positions for a pendulum. The initial positions are uniformly sampled between $\frac{3\pi}{4}$ and $\frac{5\pi}{4}$, and the target ones between $\frac{\pi}{4}$ and $-\frac{\pi}{4}$ (0 rad corresponds to the angle of the pole when it is vertical and up). The initial velocity of the pole is zero. We report success rates in Table 2. An experiment is considered successful when the pole reaches its target position within $\mp\alpha$ degrees in less than 4, 8 12 and 16s. We see that our method is capable of driving the pole to $\mp5°$ in less than 8s $58\%$ of the time, and in less than 12s $73\%$ of the time. We also propose a modified scheme where we track subgoals in the appendix (Sec. A.1).

## 5 LIMITATIONS

Although our approach is effective in controlling complex dynamical systems, it has a few limitations. The first one is that because it includes an optimal control task in the learning framework, training our model requires having optimal trajectories. We plan to gradually remove this dependency in future work. The second limitation lies in the LQR and the choice of the latent space dimension. In fact, the bottleneck in the computation of our method is the solution of LQR whose complexity is cubic in the encoding space dimension and linear with respect to the time horizon. From a practical perspective, we have not observed any limitations yet on the considered systems. But since higher dimensional systems would require higher dimensional and more complex encoding spaces, our approach might become overly expensive in this case. However LQR solvers can benefit from parallelization and thus be effective even in higher dimensions (Laine & Tomlin, 2019).

## 6 CONCLUSION

In this work, we have introduced an approach to jointly learn and control the dynamics of a nonlinear dynamical system directly from raw images. We notably leverage the Koopman formalism to learn an encoding to a space where the system dynamics are linear. To enforce the controllabilty of the system in this learned representation space, we include a control task in our learning framework to ensure the learned representation is adapted to control and we lift the control inputs to a higher dimensional space. Experimentally, we have shown that this joint strategy is effective in controlling complex dynamical systems such as pendulums and cartpoles directly from raw images. In particular, we have shown that separating the dynamics learning process from the control task leads to infeasible solutions in practice, which are avoided with the combined strategy. Next on our agenda is applying our method to conduct experiments with (simple) real robots. Future work will also include applying our approach to non-smooth dynamics or dynamics involving discontinuities.

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
