# A APPENDIX

## A.1 CONTROLLED SEQUENCES: QUALITATIVE RESULTS

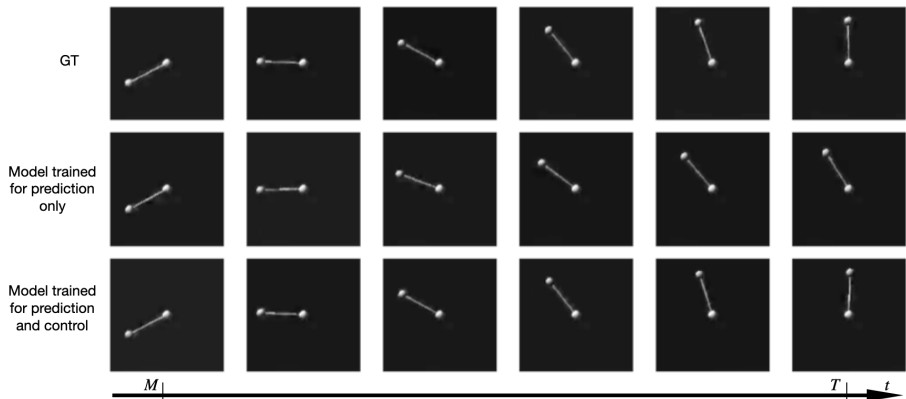

Figure 6: **Pendulum.** Video rendering of the trajectory of the controlled system. First row: the system is controlled with the ground truth optimal controls. Second row: it is controlled with control solutions to the LQR problem using the model of Bounou et al. (2021) trained for prediction only (bounou-et-al-2021). Last row: it is controlled using control solutions to the LQR problem using our model trained for prediction and control. All sequences are subsampled (we show one frame out of five). The same convention is used in Fig. 7.

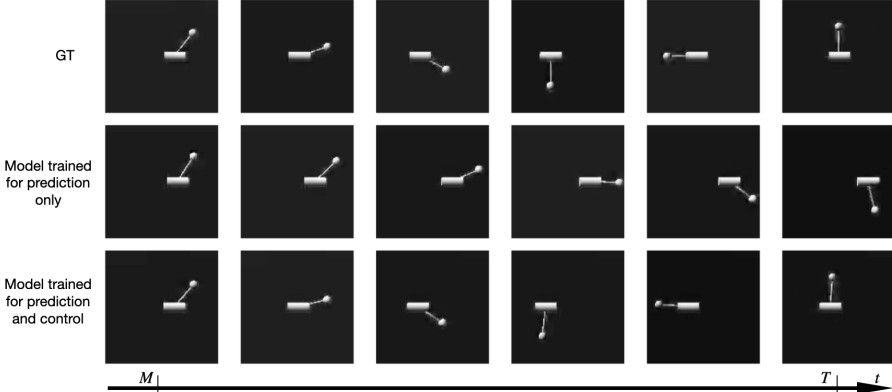

Figure 7: **Cartpole.** Video rendering of the trajectory of the controlled system.

## A.2 SUBGOAL TRACKING

Here we propose to introduce subgoals in the form of images that we assume known (but these subgoals could be discovered following the approaches of Paul et al. (2019) and Brito et al. (2021)). We solve short-horizon ($T_c = 0.4$ s) control problems to reach these subgoals, then measure and start again. Figure 8 shows that this is effective in trajectory tracking over long horizons (the ground truth trajectories are in green dashed-lines).

## A.3 ABLATION STUDY

We have run ablations on $h$ (number of consecutive frames used to form a latent code), $n$ (system encoding dimension) and $d$ (control encoding dimension) for experiments on the pendulum system. We report the RMSE on measurement (pixel) loss for both the prediction and LQR reconstruction

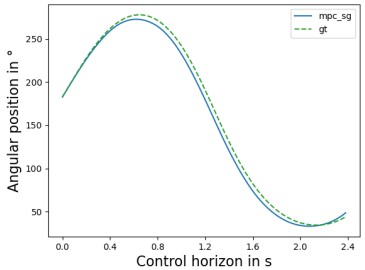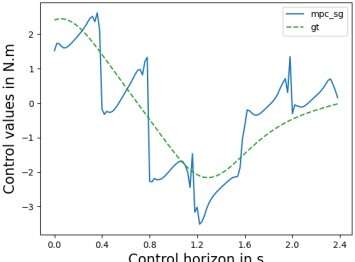

Figure 8: Trajectory tracking using subgoals every $T_c = 0.4$ s. **Left**: angular positions over time. **Right**: controls over time.

terms (terms (b) and (d) in the training objective (10) in the main paper) in Tables 3 and 4. The experiments have been run on the pendulum test dataset described in Sec. 4.1 of the main paper.

### A.3.1 HISTORY PARAMETER

Table 3: Impact of the number of history frames on the prediction and LQR reconstruction losses

|  | $h = 1$ | $h = 2$ | $h = 3$ | $h = 4$ |
|---|---|---|---|---|
| Prediction loss (b) | 0.0223 | **0.0197** | 0.0205 | 0.0224 |
| LQR reco loss (d) | 0.0203 | **0.0173** | 0.0187 | 0.0187 |

We see that using $h = 2$ helps achieve the lowest reconstruction error in prediction and control terms. This is because in the case of the pendulum, using two consecutive frames is enough to encode the velocity information, which is missing when $h = 1$.

### A.3.2 ENCODING DIMENSION

Table 4: Impact of the frame and control encoding dimensions on the prediction and LQR reconstruction losses

|  | $n = d = 2$ | $n = d = 4$ | $n = d = 8$ | $n = d = 16$ | $n = d = 32$ |
|---|---|---|---|---|---|
| Prediction loss (b) | 0.0275 | 0.0230 | **0.0197** | 0.0200 | 0.0232 |
| LQR reco loss (d) | 0.0217 | 0.0214 | **0.0173** | 0.0182 | 0.0211 |

Comparing different encoding dimensions shows that using larger ones slightly improves the prediction and control performance up to $n = d = 16$. For $n = d = 16$ and higher, training the model on more data is required to draw a conclusion. The error still remains small overall since the images in the dataset contain mainly background. Overall, the method is not too dependent on the hyperparameters.