# OpenReview forum: "Learning System Dynamics from Sensory Input under Optimal Control Principles"
_ICLR.cc/2024/Conference — Submitted to ICLR 2024_

### Official Review · Reviewer_fvtc · 2023-10-24

**Soundness:** 2 fair
**Presentation:** 3 good
**Contribution:** 2 fair
**Rating:** 3
**Confidence:** 4

**Summary:**

The authors propose a framework for learning to control systems that may have linear dynamics in a higher-dimensional space, in the vein of recent work on Koopman/DMD theory. In order to arrive at a representation that is suited for optimal control, they add an optimal control term inside the autoencoder structure, making use of part of the training data to enforce that the learned latent-space trajectory is part of an optimal trajectory in that space. They provide simulated experiments on the Cartpole and Pendulum environnments.

**Strengths:**

- The presentation is pretty good; I have no problems with the prose or structure of the text. A fairly easy read.
- The authors spend an appropriate amount of time explaining their algorithm.
- The experimental setup seems sound, and the authors provide good detail about most of the experiments
- The related work covers a good base of material from the last few years.

**Weaknesses:**

**Major**:
- The limitation acknowledged in section 5 (_"training our model requires having optimal trajectories"_) is very strong. It undermines quite a bit of the setup and background about learning from data being general, having a ton of applications, etc. Moreover, I don't believe this is even mentioned until the experiments section--it should be in the abstract.
- The experiments section is unconvincing and pretty weak. For how many thousands of learning algorithms have been thrown at the cartpole and inverted pendulum over the last 30+ years, comparing to a single algorithm is not an adequate baseline. There are many model-based and model-free approaches that have worked well on those two toy-ish problems; I'm not convinced that the authors' algorithm is required or SOTA here.
- A few experimental/implementation details have been swept under the rug -- the most prominent (unless I missed it) being the role of $T$ and $M$ -- how big do they need to be in theory, in practice, how do you set them.
- No code was provided (as far as I can tell in the reviewer console).

**Minor**:
- Table 1 is not a good way to present this data (which isn't compared to the other alg.), the tolerances seem pretty loose as well.
- I have no idea what Table 2 and the related section are talking about. It seems like it is somehow implementing an MPC-style algorithm, but it's really unclear.
- The related work doesn't acknowledge almost anything written from about 1965 to 2005, and only slightly more pre-2015. The words "system identification" do not appear in the paper at all, which is concerning.

**Questions:**

- How does this algorithm perform on linear systems? It seems like it should do the right thing only if _optimal_ trajectories are given, which is the setup acknowledged in your paper, but points toward it not doing the right thing for linear systems if you have general data.
- How are $M$ and $T$ determined?

---

### Official Review · Reviewer_ianP · 2023-10-30

**Soundness:** 3 good
**Presentation:** 3 good
**Contribution:** 2 fair
**Rating:** 5
**Confidence:** 4

**Summary:**

This paper looks to solve challenging nonlinear control problems from raw images with the help of the Koopman formalism. By "lifting" the sensor and control variables into a different space (embedding), one where the underlying dynamics are linear, LQR is able to achieve the control objective. A network is set up that involves encoding the sensor and control signals, fitting a linear system to these, solving for a feedback control law with LQR and then decoding the results back to sensors and controls. This network is trained end to end with a loss function that penalizes autoencoder/decoder errors, prediction errors, controllability, and control usage.

**Strengths:**

1. Using DNN's to "learn" a good koopman embedding from images is a great idea, removing much of the legwork involved in Koopman control, as well as helping it generalize to arbitrary robotic systems.
2. The structure of the network is clearly laid out, and each component is logically sound. By giving each portion of the network a section, each component of the network is well defined.
3. All of the notation is clearly labeled and easy to follow.
4. The section on LQR with "delayed" coordinates is helpful and clear.
4. The related work section is thorough and helpful for those less familiar with this line of work.

**Weaknesses:**

1. The impact of this approach may be significant, but the scope of the demonstrations in the paper is so narrow that I am unconvinced. Since they only look at 2 very simple systems (pendulum and cartpole), and rely on "demonstrations" from trajectory optimization, there is no evidence that this approach will work well for other systems. Even having 2-3 more toy problems would have greatly improved the argument for generalizability. I feel that the most exciting part of this paper is the ability to deal with complex robotic systems at the image level, but it was only shown to work on simple toy problems with many readily available solution methods.
2. The results are compared to Bounou et al. 2021, which makes sense because the approaches are so similar. However, since you are only considering deterministic systems with computer generated images trained on a dataset of optimized trajectories, a comparison to a simple supervised learning regression based controller would have been very useful. This comparison would (hopefully) have shown that a simple supervised learning approach was ineffective.

**Questions:**

1. The plots in Figure 5 for the cartpole look abnormal to me. Is the control for this system the torque at the base of the pole? If so, this is not the classic cartpole, and there should be a note describing this explicitly.
2. The decision to use proximal method of multipliers for solving the least squares problem in equation 6 is very interesting. What is wrong with solving the L2 regularized version? A regularizer seems like it could be helpful if the new Koopman state is slightly "over parametrized". Also, why proximal method of multipliers instead of something more straightforward like iterative refinement? I am not suggesting the chosen method is bad, I would have just appreciated another sentence or two justifying it.

---

### Official Review · Reviewer_6hvE · 2023-11-01

**Soundness:** 2 fair
**Presentation:** 2 fair
**Contribution:** 1 poor
**Rating:** 1
**Confidence:** 5

**Summary:**

The paper using Koopman operator theory (KOT) to transform a nonlinear system to a lifted space linear system. This linear system is then used for LQR control design for stabilising the original nonlinear system. Numerical simulations on pendulum and cartpole are performed to verify their claims.

**Strengths:**

1) The authors have provided sufficient details for reliplicating their simulations

2) The paper is well written which makes it easier for readers to understand the nuances and its contributions

**Weaknesses:**

I've had the opportunity to delve into Koopman operator theory (KOT) in my past research so I am very well aware of the nuances in this paper. The weakness of the paper are as follows:

1) It must be noted that that control-affine systems transform to a bilinear control Koopman based system (of Koopman based linear system) in the lifted space under particular conditions, as exemplified by Theorem II.1 in the paper [1]. It's also worth mentioning that these bilinear/linear forms may not be universally applicable to all general nonlinear systems, especially those that do not adopt the control affine form. Some recent works on KOT have also showcased that a general control-affine nonlinear system can be transitioned to a more generalized input-separable Koopman system, with bilinear and linear forms being special instances of these separable Koopman forms.

2) Note that KOT based approximation is only valid for control affine systems and not general nonlinear systems of form (1) given in the paper.

3) The problem of control of dynamical systems using Koopman operator theory using MPC [4], LQR[5], Lyapunov based[6,7] methods has already been explored before. Furthermore, [5] considers more complicated nonlinear systems such as lunar lander in their simulations. There are more subsequent works on Koopman based control applied to real world applications such as soft robots [8].

4) On the topic of stability in Koopman-based learned matrices you use the proximal method of multipliers as in eqn (6) of your paper, however there are several contributions, such as [3], and its subsequent related studies that have already addressed this issues. However, you have not cited them in the literature. I believe you are just using another approach to promote stability.

5) Considering the above points, a more comprehensive literature review on the Koopman operator for controlled/non-controlled dynamical systems might enhance the paper's breadth. Furthermore, you should consider examples such as quadrotors, vehicle dynamics based on bicycle model etc.which are more nonlinear and applicable to real world.

6) The training objective is very standard in KOT literature and does not solve any new challenges. Furthermore, I feel that other linearization based methods such as the iterative LQR (iLQR) would work better or equivalent to the proposed Koopman based LQR approach.

7) Extensive numerical comparisons with well known and established control techiques are missing in their paper such as Nonlinear MPC, iLQR, PID and recent methods like MPPI and so on.

[1] Bruder, Daniel, Xun Fu, and Ram Vasudevan. "Advantages of bilinear Koopman realizations for the modeling and control of systems with unknown dynamics." IEEE Robotics and Automation Letters 6, no. 3 (2021): 4369-4376.

[2] Lusch, Bethany, J. Nathan Kutz, and Steven L. Brunton. "Deep learning for universal linear embeddings of nonlinear dynamics." Nature communications 9, no. 1 (2018): 4950.

[3] Fan, Fletcher, Bowen Yi, David Rye, Guodong Shi, and Ian R. Manchester. "Learning stable Koopman embeddings." In 2022 American Control Conference (ACC), pp. 2742-2747. IEEE, 2022.

[4] Korda, Milan, and Igor Mezić. "Linear predictors for nonlinear dynamical systems: Koopman operator meets model predictive control." Automatica 93 (2018): 149-160.

[5] Han, Yiqiang, Wenjian Hao, and Umesh Vaidya. "Deep learning of Koopman representation for control." In 2020 59th IEEE Conference on Decision and Control (CDC), pp. 1890-1895. IEEE, 2020.

[6] Zinage, Vrushabh, and Efstathios Bakolas. "Neural koopman lyapunov control." Neurocomputing 527 (2023): 174-183.

[7] Huang, Bowen, Xu Ma, and Umesh Vaidya. "Feedback stabilization using Koopman operator." In 2018 IEEE Conference on Decision and Control (CDC), pp. 6434-6439. IEEE, 2018.

[8] Bruder, Daniel, Xun Fu, R. Brent Gillespie, C. David Remy, and Ram Vasudevan. "Data-driven control of soft robots using Koopman operator theory." IEEE Transactions on Robotics 37, no. 3 (2020): 948-961.

**Questions:**

1) It would be beneficial more general nonlinear examples such as quadrotors, vanderpol oscillators or vehicle dynamics based on bicycle model.

2) The nonlinear system given in (1) must not be general to use KOT. The Koopman based lifted space approximation only works for "control -affine" nonlinear systems.

3) The authors have not cited sufficient Koopman related control papers in literature.

---

### Official Review · Reviewer_wihP · 2023-11-02

**Soundness:** 2 fair
**Presentation:** 2 fair
**Contribution:** 2 fair
**Rating:** 5
**Confidence:** 3

**Summary:**

This paper aims to identify the underlying dynamics driving physical systems, which involves learning to incorporate states and controls into a feature space where the dynamics are linear. The proposed method is validated through simulation experiments using nonlinear dynamical systems and demonstrated its effectiveness in driving physical systems and predicting future states. The main contribution is a new method for learning system dynamics from sensory information under optimal control principles, which can be applied to physical systems with nonlinear dynamics.

**Strengths:**

The strength includes the investigation of how to incorporate states and controls into a dynamically linear feature space. The proposed method is validated through simulation experiments using nonlinear dynamical systems.

**Weaknesses:**

The computational difficulty is the solution of the LQR, which may become too expensive for high-dimensional systems. Also,  the proposed method is not explicitly compared with other existing methods.

**Questions:**

Please present detailed comparison studies with existing approaches.

---

### Meta-Review · Area_Chair_RUR3 · 2023-12-09

**Metareview:**

Synopsis: The paper presents an approach to learning system dynamic from noisy observations by learning embeddings of the states and controls in a space where the transition function is linear. The paper includes results from simulation.

Strengths:
+ Well motivated, important problem that occurs with real systems

Weaknesses:
- Questions on technical soundness and unstated assumptions (please see review 6hvE)
- Empirical results not thorough, and omit ready baselines with supervised learning

**Justification For Why Not Higher Score:**

The reviews are clear - unfortunately only one of the reviewers delved into the technical approach in detail, but this review raised some valid and good technical concerns. The other reviews were universally underwhelmed by the results and the omission of obvious baselines.

**Justification For Why Not Lower Score:**

N/A

---

### Decision · Program_Chairs · 2024-01-16

Reject